# Stimulus-choice (mis)alignment in primate area MT

**Yuan Zhao**[1], **Jacob L. Yates**[2], **Aaron J. Levi**[3], **Alexander C. Huk**[3], **Il Memming Park**[1] *

**1** Department of Neurobiology and Behavior, Stony Brook University, Stony Brook, New York, United States of America, **2** Brain and Cognitive Science, University of Rochester, Rochester, New York, United States of America, **3** Center for Perceptual Systems, Departments of Neuroscience & Psychology, The University of Texas at Austin, Austin, Texas, United States of America

* memming.park@stonybrook.edu

**Data Availability Statement:** All data and code can be reached at https://doi.org/10.6084/m9.figshare. 11413182.v1.

**Funding:** YZ, ACH and IMP were supported by NSF IIS-1734910. National Science Foundation, www.

## Abstract

For stimuli near perceptual threshold, the trial-by-trial activity of single neurons in many sensory areas is correlated with the animal's perceptual report. This phenomenon has often been attributed to feedforward readout of the neural activity by the downstream decision-making circuits. The interpretation of choice-correlated activity is quite ambiguous, but its meaning can be better understood in the light of population-wide correlations among sensory neurons. Using a statistical nonlinear dimensionality reduction technique on single-trial ensemble recordings from the middle temporal (MT) area during perceptual-decision-making, we extracted low-dimensional latent factors that captured the population-wide fluctuations. We dissected the particular contributions of sensory-driven versus choice-correlated activity in the low-dimensional population code. We found that the latent factors strongly encoded the direction of the stimulus in single dimension with a temporal signature similar to that of single MT neurons. If the downstream circuit were optimally utilizing this information, choice-correlated signals should be aligned with this stimulus encoding dimension. Surprisingly, we found that a large component of the choice information resides in the subspace orthogonal to the stimulus representation inconsistent with the optimal readout view. This misaligned choice information allows the feedforward sensory information to coexist with the decision-making process. The time course of these signals suggest that this misaligned contribution likely is feedback from the downstream areas. We hypothesize that this non-corrupting choice-correlated feedback might be related to learning or reinforcing sensory-motor relations in the sensory population.

## Author summary

In sensorimotor decision-making, internal representation of sensory stimuli is utilized for the generation of appropriate behavior for the context. Therefore, the correlation between variability in sensory neurons and perceptual decisions is sometimes explained by a causal, feedforward role of sensory noise in behavior. However, this correlation could also originate via feedback from decision-making mechanisms downstream of the sensory representation. This cannot be resolved by analyzing single unit responses, but requires a

nsf.gov. ACH was supported by NEI EY017366. National Eye Institute, nei.nih.gov. JLY and AJL were supported by National Institutes of Health under Ruth L. Kirschstein National Research Service Awards T32DA018926 from the National Institute on Drug Abuse and T3EY021462 from the National Eye Institute. JLY was supported by T32EY007125 from the National Eye Institute. JLY is an Open Philanthropy fellow of the Life Sciences Research Foundation. The funders had no role in study design, data collection and analysis, decision to publish, or preparation of the manuscript.

**Competing interests:** The authors have declared that no competing interests exist.

population level analysis. Area MT contains both sensory and choice information and is known to be the key sensory area for visual motion perception. Thus the decision-making process may be corrupting the sensory representation. However, we find that the sensory stimuli and choice variables are separate at the population level, contradicting the previous interpretations based on single unit recordings. This new insight postulates how neural systems can maintain a mixed representation while allows learning and adaptation.

## Introduction

Sensory cortical neurons exhibit substantial variability to repeated presentations of the same stimulus [1, 2]. This variability depends on the specifics of the sensory stimulus and task being performed [3–7], and is often correlated with the trial-by-trial perceptual report of the animal [8–11]. This trial-by-trial correlation between neural responses and perceptual reports, often quantified as choice probability (CP), has long been of interest for its potential to reveal the mechanisms by which downstream areas read out the response of relevant population of sensory neurons [12–14]. However, this interpretation is complicated by the presence of interneuronal correlations [15], top-down feedback [9, 16] and also depends on assumptions about the readout mechanisms of downstream brain areas [12, 14, 16, 17].

Several models of perceptual decision-making have been proposed to explain the empirical relationships between stimuli, neural responses, and behavioral choices [12, 14, 16]. Existing proposals come in two basic flavors: those that posit an optimal readout that is limited by shared neural variability [14, 18, 19] and those that assert that choice-related feedback modifies the signals in sensory areas [16, 20]. Several recent experimental results support the feedback hypothesis [7, 9, 20]. Although feedback can be interpreted in terms of probabilistic inference [16], the resulting pattern of variability in sensory areas will reduce the information about the stimulus [16, 19, 21] and impair performance on the task [20]. Why would the brain bother to feedback a choice or decision that corrupts the sensory information and make it do worse on the task? Here, we propose an alternative hypothesis: the feedback can be non-corrupting, effectively multiplexing choice signals in a sensory population without diminishing information about the stimulus.

To visualize the space of hypotheses and how they can be distinguished, it is helpful to summarize the joint activity of a population of neurons with respect to the stimulus driven activity. Fig 1 demonstrates this alignment conceptually and the effect of each type of choice models in this space. Specifically, for a population of only two neurons, the joint activity of the population can be represented as point clouds in a 2-dimensional space where each axis represents an individual neuron's activity (Fig 1A). For a 1-dimensional stimulus (as is typically used in discrimination paradigms), different values of the stimulus (red and black) drives activity that falls along a 1-dimensional "stimulus axis". For simplicity, in this toy example, we assume there is no structure in the co-variability that can be exploited for decoding. In this case, increased variability along the stimulus axis (the so-called information-limiting noise) will change the amount of information about the stimulus, while, importantly, variability orthogonal to the stimulus axis will not [19, 22–25]. We call this variability direction the "non-stimulus axis" (Fig 1A). In larger populations, the stimulus may reside in a subspace of higher dimension, however, we can use statistical classification methods to determine the stimulus subspace and non-stimulus subspaces in general.

By realigning the population activity to the "stimulus axis", the effect of noise correlations and feedback can be visualized clearly. Noise correlation (or trial-to-trial co-variability) is the

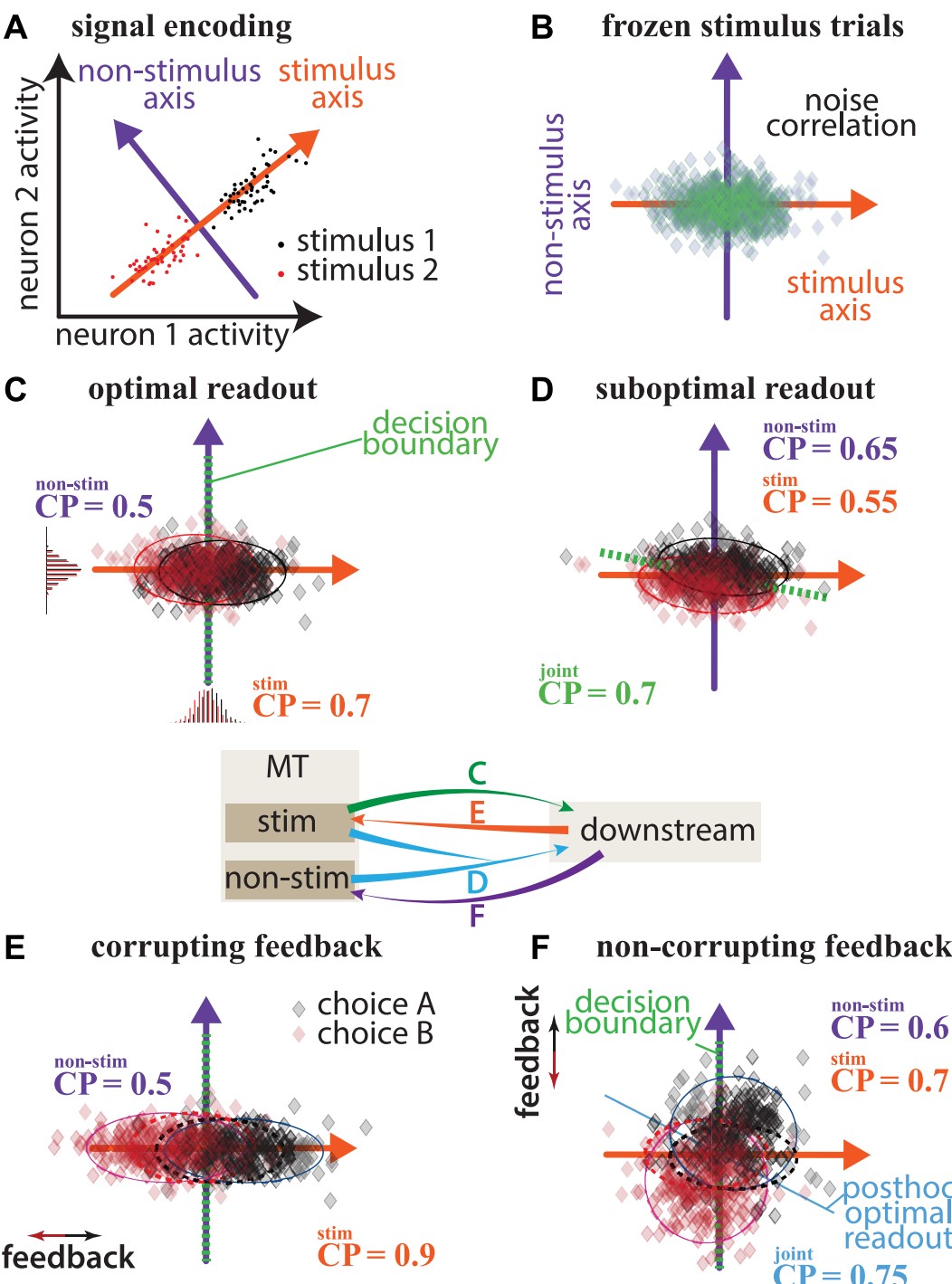

**Fig 1. Hypotheses on the sources of choice correlations in sensory area.** (A) Joint activity of the population. The point cloud represents neuronal activities colored by stimulus direction. The neural space can be divided into stimulus and non-stimulus axes. (B) Noise correlation is any elongation of the joint activity point cloud for repeats of the same stimulus. (C) Optimal readout. The optimal decision boundary is a criterion line orthogonal to the stimulus axis. All CP is due to readout and there is no CP in the non-stimulus axis. (D) Suboptimal readout. The decision boundary is not orthogonal to the stimulus axis. CP exist in both axes. (E) Corrupting feedback. The choice is fed back and pushes variability along the stimulus axis. This increases CP along the stimulus axis without affecting the non-stimulus axis, and causes more variability along the stimulus axis. (F) Non-corrupting feedback. Feedback pushes choice information in the non-stimulus axis and increases CP in the non-stimulus axis without adding CP in the stimulus axis.

joint activity distribution for repeats of the same stimulus (Fig 1B). The downstream decision making process may solely rely on the MT activity on the stimulus-axis to generate behaviors or may also utilize the signal from the non-stimulus axis as well. Meanwhile, the downstream behavior-relevant neural activity that builds up over time can be fed back to MT on either axis. The combination gives rise to the following four hypotheses (corresponding to Fig 1C, 1D, 1E and 1F):

1. The classic feedforward hypothesis: the decision process optimally reads out MT population, and noise correlation in MT induces CP greater than chance (i.e., 0.5) along the stimulus axis. In this model, all CP in MT is due to readout and there is no CP in the non-stimulus axis.

2. The suboptimal readout hypothesis: an alternative feedforward mechanism that reads out MT population information such that it inherits the variability in the non-stimulus axis giving rise to above-chance CP in the non-stimulus axis (Fig 1D).

3. The corrupting feedback hypothesis [16, 20, 21]: the choice is fed back along the stimulus axis. If the feedback is positively signed, this increases the measured CP and causes more variability along the stimulus axis without affecting the non-stimulus axis, and may bias the performance on the task for weak stimuli (Fig 1E).

4. The non-corrupting feedback hypothesis: feedback could avoid interfering with the stimulus representation by pushing choice information only in the non-stimulus axis (Fig 1F). This increases CP in the non-stimulus axis without adding CP in the stimulus axis and does not influence the optimal stimulus readout.

To test these different hypotheses requires an analysis of the joint statistics of populations of sensory neurons while subjects perform a discrimination task. Here, we apply the recent developments in statistical dimensionality reduction of single-trial population recordings [26] to examine how information about the stimulus and choice are encoded jointly in small populations of simultaneously recorded MT neurons during perceptual reports about integrated motion direction [27]. The effects of stimulus, choice, and trial-to-trial variability present in the population activity are decomposed into shared low-dimensional latent factors and noise that is private to each neuron. Unsurprisingly, low-dimensional shared signals capture a majority of the variability in these data as seen previously in other areas [6, 26, 28, 29]. By aligning the latent signals to the stimulus and task variables, we were able to investigate how stimulus and choice are encoded by neurons collectively.

We found that the task variable (visual motion) was primarily captured by a single latent factor, indicating that the high-dimensional visual stimulus was represented in a low-dimensional, task-relevant manner across the MT population. Additionally, we found that the choice-correlated variability in the population was mainly captured by the latent subspace orthogonal to the task dimension and appears slowly during the stimulus presentation. These results suggest the choice signal is fed back to sensory cortex in the null space of the stimulus—multiplexing choice signals in sensory areas without corrupting information about the stimulus. This feedback signal could be critical for adapting sensory representations while learning new tasks or in non-stationary environments [30, 31].

## Materials and methods

### Electrophysiology, task, and behavioral data

Data were recorded from three adult rhesus macaque monkeys (two males, P & L, and one female, N) performing perceptual decision-making task for multiple sessions (P: 9, L: 13, N:

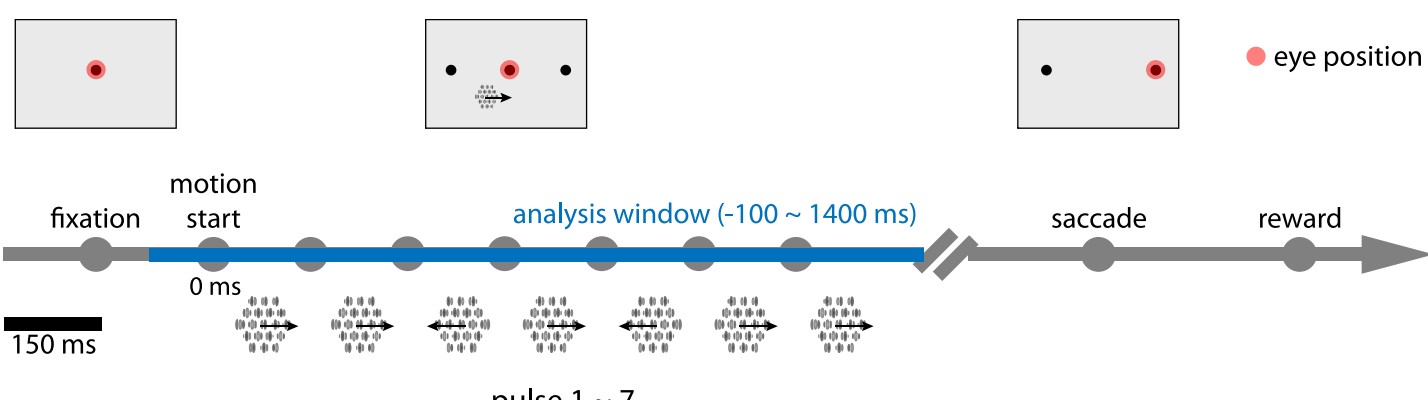

**Fig 2. Experimental setup: Motion discrimination task.** Trials started with a fixation at the center of the monitor. 7 consecutive motion pulses were presented to the monkey while monkeys hold the fixation. Each motion pulse consists of drifting and flickering Gabor patches and lasts 150 ms. Random, signed motion strength was determined by changing the proportion of drifting vs. flickering Gabors patches. Monkeys reported their choice depending on the net direction by making a saccade to one of the two choice targets after the fixation point to disappear.

10) as reported in [27, 32] (with additional sessions added). Spike trains from area MT were obtained via linear electrode arrays. All procedures were performed in accordance with US National Institutes of Health guidelines, were approved by The University of Texas at Austin Institutional Animal Care and Use Committee. Briefly, each trial began with a fixation at the center of the monitor. Then the visual stimuli were presented to the monkey. The monkey must hold fixation during the presentation (Fig 2). In each trial, 7 consecutive motion pulses were presented to the monkeys, each lasting 150 ms. Each motion pulse consisted of a hexagonal grid of drifting or flickering Gabor patches. The strength (controlled by the number of drifting patches) and direction of each pulse was randomly drawn from Gaussian distribution and rounded. The monkeys made a saccade to one of the two targets after the fixation point disappear. Rewards were given for making a correct choice if the total sum of motion pulses was greater in the corresponding direction or at random with probability 0.5 on the zero sum trials. We keep the recordings from 100 ms before the visual stimuli onset to 350 ms after the visual stimuli offset. We analyze sessions with at least 10 neurons in order to extract latent factors (for a total of 14 sessions). Length of sessions ranged from 245 to 1000 good trials.

## Single-trial latent dynamics of population

To understand how stimulus and perceptual choice are encoded across the population, we employed the variational latent Gaussian process (vLGP) method [26] to extract single-trial low-dimensional latent factors from population recordings in area MT. We used the recording of the period from 100 ms before stimulus onset to 350 ms after offset, and binned the spike counts at 1 ms resolutions. Let $\mathbf{x}_k$ denote the $k$-th dimension of the latent factors. We assumed that the spatial dimensions of latent factors are independent and imposed a Gaussian Process (GP) prior to the temporal correlation of each dimension,

$$\mathbf{x}_k \sim \mathcal{N}(0, \mathbf{K}). \tag{1}$$

To obtain smoothness, we used the squared exponential covariance function and respective covariance matrix $\mathbf{K}$ in the case of discrete time. Let $y_{tn}$ denote the occurrence of a spike of the $n$th neuron at time $t$, $y_{tn} = 1$ if there was a spike at time $t$ and $y_{tn} = 0$ otherwise at this time resolution. Then $\mathbf{y}_t$ is the vector of length $N$, total number of neurons in a session, that concatenates all neurons at time $t$. The spikes $\mathbf{y}_t$ are assumed to be a point-process generated by the

latent state $\mathbf{x}_t$ at that time via a linear-nonlinear model,

$$\mathbf{y}_t \sim \text{Poisson}(\exp(\mathbf{A}\mathbf{x}_t + \mathbf{b})). \tag{2}$$

To infer the latent factors ($\mathbf{x}_t$ for each trial) and the model parameters ($\mathbf{A}$ and $\mathbf{b}$), we used variational inference technique, as the pair of prior and likelihood do not have an tractable posterior. We assumed parametric variational posterior distribution of the latent factors,

$$q(\mathbf{x}_k) = \mathcal{N}(\boldsymbol{\mu}_k, \boldsymbol{\Sigma}_k). \tag{3}$$

We analyze the mean $\{\boldsymbol{\mu}_k\}$ as the latent factors in this study. The detail of inference is described in the supplementary materials S1 Text. The dimensionality of the latent factors was determined to be 4 by leave-one-neuron-out cross-validation on the session with the largest population (3). All the sessions with more than 10 simultaneously recorded units were included in this study.

## Pulse-triggered average

To measure the relationship between the time-varying pulse strength and the inferred latent factors, we measured the contribution of pulses to the latent factors. The pulse-triggered average (PTA) measures the change in latent factors resulting from an additional pulse at a particular time of unit strength. To calculate the PTA, we used the pulse stimulus and latent response at 1 ms resolution. For each session, let $s_i$ denote the value of the $i$-th motion stimulus, and let $x_{tk}$ denote the $k$-th dimension of the latent factors at time $t$. All trials were concatenated such that the latent factors $\mathbf{X}$ is a matrix of length $T \times 4$, where $T$ is the total time. For the $i$-th pulse, $s_i$ is the number of Gabors pulsing, with $s_i > 0$ for pulses in one direction and $s_i < 0$ for pulses in the other direction. To calculate the temporal lags of the PTA, we built design matrices, $\mathbf{D} = [\mathbf{D}_1, \mathbf{D}_2, \ldots, \mathbf{D}_7]$. For the $i$-th pulse, the design matrix $\mathbf{D}_i$ is a $T \times 28$ matrix that consists of 4 cosine basis functions at the $4i + 1$, $4i + 2$, ..., $4i + 4$-th columns and 0 elsewhere. These basis functions starts at 0 ms, 50 ms, 100 ms and 150 ms after the onset, lasts 100 ms each and spans the rows of $\mathbf{D}_i$. The magnitude of the bases is equal to the corresponding pulse value $s_i$. We calculated a separate $\mathbf{D}_i$ for each of the seven pulses and concatenated them to obtain a design matrix for all seven pulses and estimated the weights with $\ell_2$-regularization,

$$\mathbf{X} = \mathbf{D}\mathbf{W} + \mathbf{E}$$
$$\mathbf{W} = \underset{\mathbf{W}}{\arg\min} \|\mathbf{X} - \mathbf{D}\mathbf{W}\|_2^2 + \gamma\|\mathbf{W}\|_2^2 \tag{4}$$

where $\mathbf{W}$ is the weight matrix to estimate and $\mathbf{E}$ is the Gaussian noise matrix and the regularization hyperparameter $\gamma$ was chosen by the generalized cross-validation (GCV) [33]. The PTA was calculated with the design matrices of unit-strength pulse and the estimated weights $\mathbf{W}$. We smoothed the PTA with a temporal Gaussian kernel (40 ms kernel width).

Subject to arbitrary rotations, a latent trajectory forms an equivalence class of which the members have the same explanatory power in the vLGP model. We seek a particular rotation for each session that makes the encoded task signal concentrate in the first few dimensions. By singular value decomposition, $\mathbf{W}^\top = \mathbf{U}\mathbf{S}\mathbf{V}^\top$, we rotate the factors $\mathbf{x}$ to $\mathbf{U}^\top\mathbf{x}$.

## Choice decoder

Since there were some recording sessions with less than ideal number of frozen trials (identical visual motion trials) for the calculation of choice probability, we instead analyzed the "weak" trials of which the monkeys' correct rate was below a threshold (65%). We started at the trials

of zero pulse coherence and gradually increased the magnitude of coherence (absolute value) until the correct rate reached the threshold. One of the sessions containing less than 100 weak trials was excluded in this analysis.

We removed the stimulus directions that are encoded in the latent factors and raw population activity of weak trials by regressing out the pulses and analyzed the residuals. The latent factors and population activity were re-binned at 100 ms resolution where the value of each bin is the sum of latent state $\mathbf{x}_t$ or spike counts $\mathbf{y}_t$ over the bin for $t = 1, 2, \ldots, T$. For each $t$, we assumed a linear model to predict its value

$$\mathbf{x}_t = \sum_{i=1}^{7} \mathbf{w}_{ti} s_i + \mathbf{e}, \tag{5}$$

where $s_i$ denote the strength of the $i$-th pulse, $\mathbf{w}_{ti}$ is the weight vector corresponding to the bin and pulse, and $\mathbf{e}$ is the homogeneous Gaussian noise across all bins. We estimated the weight vector by least-squares with $\ell_2$-regularization to prevent over-fitting,

$$\mathbf{w}_{ti} = \arg \min_{\mathbf{w}_{ti}} \left\| \mathbf{x}_t - \sum_{i=1}^{7} \mathbf{w}_{ti} s_i \right\|_2^2 + \gamma \| \mathbf{w}_{ti} \|_2^2. \tag{6}$$

Again, the hyperparameter of regularization was chosen by GCV. For the raw population activity, we did the same regression, replacing $\mathbf{x}_t$ with the spike count $\mathbf{y}_t$. We then analyzed the contribution of behavioral choice on the residuals

$$\mathbf{r}_t = \mathbf{x}_t - \sum_{i=1}^{7} \mathbf{w}_{ti} s_i. \tag{7}$$

For the whole trial we used the sum residual of the windows $\mathbf{r} = \sum_t \mathbf{r}_t$. The range of $t$ depends on the period of interest.

We trained logistic models, to which we refer to as *choice decoders*, to predict the subject's choice on each trial using either latent factors or population responses. The weights $\boldsymbol{\beta}$ and bias $\beta_0$ were estimated by maximum likelihood with $\ell_2$-regularization,

$$\boldsymbol{\beta}, \beta_0 = \arg \max_{\boldsymbol{\beta}, \beta_0} \log L(\text{choice} \mid \mathbf{r}; \boldsymbol{\beta}, \beta_0) - \gamma \| \boldsymbol{\beta}, \beta_0 \|_2^2. \tag{8}$$

Due to small sample sizes, the hyperparameter of regularization was chosen via 3-fold stratified (balanced classes in test set) cross-validation for every session individually.

## Choice mapping

The conventional choice probability only applies to univariate variables. However, either the latent factors or population activity is multivariate. We transformed the multivariate variables mentioned above onto a one-dimensional subspace that has the same direction as the choice through the choice decoders,

$$c = \frac{1}{1 + e^{-\boldsymbol{\beta}^\top \mathbf{r} - \beta_0}} \tag{9}$$

We refer to the transform as the *choice mapping*. The quantity $c$ is a normalized value within [0, 1] that maps the residual onto the choice direction [34], and enables aggregation across sessions as well.

In order to prevent potential inflation of choice probability due to high dimensionality (3D), we did not only regularized the choice decoder to estimate the weights but also use the

choice mapped values of the test set (pooled samples held-out by cross-validation). This approach guarantees that the overfitting of choice decoder will not result in overestimating the choice probability. The synthetic example (S1 Fig) also verified that adding choice-irrelevant dimensions does not inflate the choice probability.

We pooled these mapped values and aggregate them across all sessions. By plugging different dimensions of latent factors or population activity as **r** in the mapping, we obtained the choice-mapped values of the stimulus-dimension, non-stimulus-dimensions of latent factors and the whole population. With these mapped values, we calculated the choice probability of the corresponding dimensions.

To investigate the effect of different dimensions on the choice, we did sequential likelihood ratio tests through adding the choice-mapped value of stimulus-dimension, non-stimulus-dimensions and the population one by one to a logistic model that predicts the choice,

$$
\begin{aligned}
LR_1 &= \frac{L(\text{choice} \mid c_{\text{stimulus}})}{L(\text{choice} \mid c_{\text{stimulus}}, c_{\text{non-stimulus}})} \\
LR_2 &= \frac{L(\text{choice} \mid c_{\text{stimulus}}, c_{\text{non-stimulus}})}{L(\text{choice} \mid c_{\text{stimulus}}, c_{\text{non-stimulus}}, c_{\text{population}})}
\end{aligned}
\tag{10}
$$

where the values of $c_{\text{stimulus}}$, $c_{\text{non-stimulus}}$ and $c_{\text{population}}$ were calculated by Eq 9 with the response **r** (Eq 7) and weights $\{\beta_0, \boldsymbol{\beta}\}$ estimated via Eq 8 of corresponding axes.

To investigate the time course of choice probabilities, we used choice decoders to perform choice-mapping on the whole data with a 100 ms non-overlapping moving window. The choice decoders were fitted to early (200–500 ms), middle (600–900 ms) and late (1000–1300 ms) periods of non-stimulus latent factors, and regularized with cross-validation mentioned above. The choice probabilities of all time bins were then calculated on the choice-mapping using the three decoders individually.

## Results

### Low-dimensional shared variability structure

Three monkeys performed a motion-pulse direction discrimination task with an eye movement to one of two targets [32]. The visual stimulus was presented as a sequence of 7 temporally coherent motion pulses of varying strength. An ensemble of MT neurons were simultaneously recorded using multi-electrode arrays. Given the recording, we statistically infer a low-dimensional latent factors that explains the shared component of the high-dimensional variations in the observed spiking activity. Conventional analysis methods such as factor analysis or principal component analysis assume either observation models inappropriate for spikes (e.g. Gaussian) or linear dynamics that lack expressive power to describe any non-trivial computation. To overcome these disadvantages, we imposed a general (nonlinear) Gaussian process prior on the latent factors and assumed a point-process observation model to account for spikes. The generative model was fit using the variation latent Gaussian process (vLGP) method to recover nonlinear smooth latent factors from population recordings [26]. Fig 3A shows the scheme of the model and an example trial. The population firing rates are driven by the latent factors through a linear-nonlinear cascade. The loading matrix linearly maps the high-dimensional observation space to the low-dimensional latent space, of which the rows corresponding to the neurons and the columns corresponding to the latent factor dimensions. The extracted latent factors captured the shared variability of the population activity, while the individual variability of each neuron was explained by stochastic generation of spike trains. The dimensionality of latent factors was chosen to be 4 by a leave-one-out cross-validation

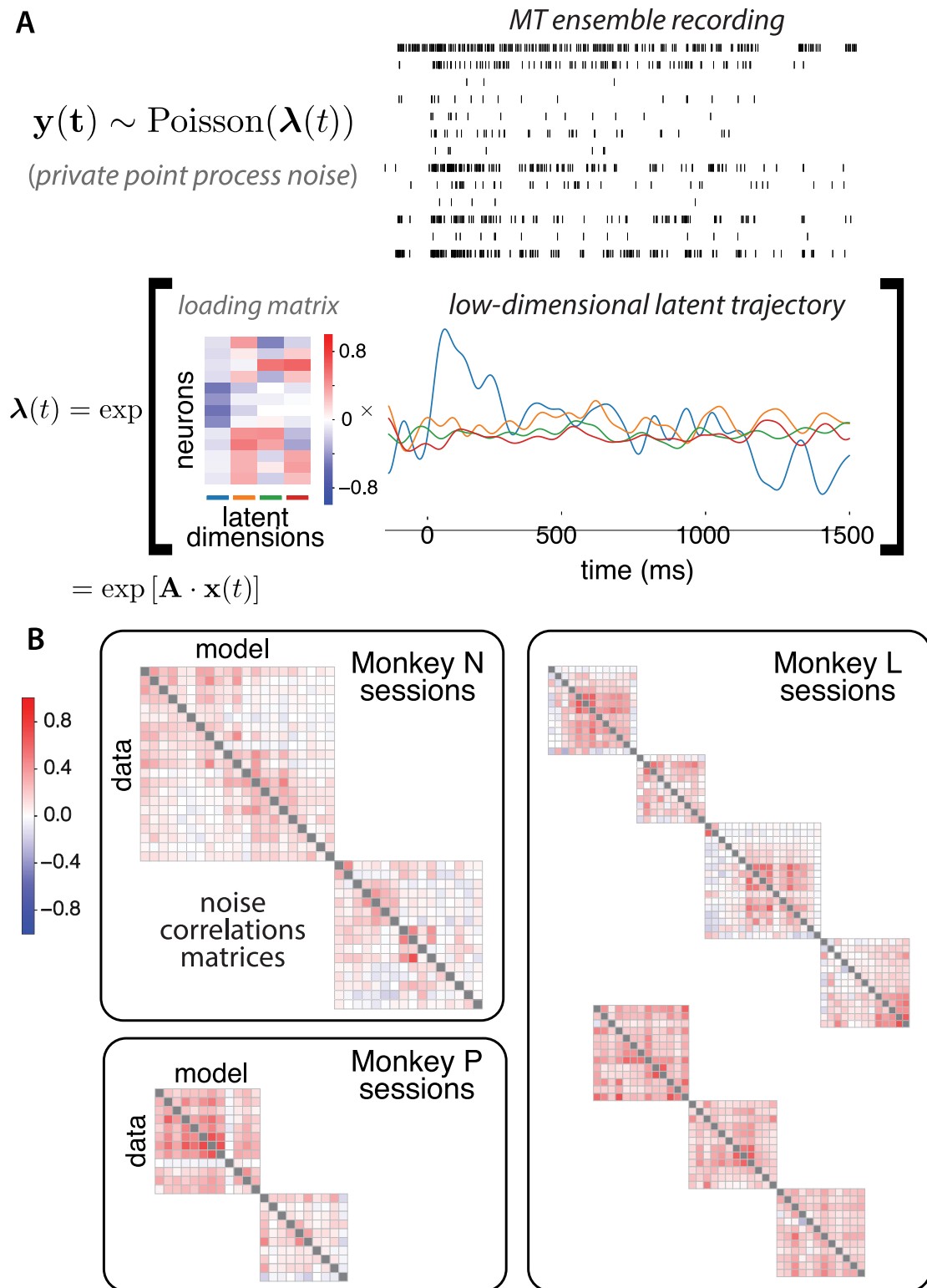

**Fig 3. Probabilistic description of a single trial using variational latent Gaussian process method and resulting noise correlation.** (A; top) Simultaneously recorded spike trains of the MT units in an example trial aligned to stimulus onset ($\mathbf{y}_t$ in Eq 2). (A; bottom) Corresponding 4-dimensional factors. The rank-4 matrix multiplication of the loading matrix (matrix $\mathbf{A}$ in Eq (2)), and latent factors are exponentiated to produce the population rate. The loading matrix is rotated to maximize stimulus encoding (see Fig 4), so that the first column has the strongest stimulus response. The inferred latent factors ($\mathbf{x}_t$ in Eq (2)) are colored to

indicate the respective factors corresponding to the loading matrix. (B) The pairwise noise correlation matrices (neuron by neuron) for the sessions with frozen trials (trials with identical stimulus). The lower triangles are the correlations calculated from the raw data, and the upper triangles are the correlations from the reconstruction by the inferred 4-dimensional latent factors. Time bin size 100 ms.

scheme on the session with the largest population ($N = 21$ neurons). To aggregate analysis across sessions, we fixed this dimensionality of the latent factors.

To validate the model, we evaluate the pairwise noise correlations between neurons on randomly interleaved frozen trials where the stimulus was held constant (Fig 3B). With the inferred latent factors and loading matrix, we can generate spike trains from the model. We calculated the noise correlation matrices from data and reconstructed spikes. To quantify the goodness of the model capturing the noise correlation, we calculated the $R^2 \doteq 1 - \|C_{\mathrm{data}} - C_{\mathrm{model}}\|_F^2 / \|C_{\mathrm{data}}\|_F^2$ where $C_{(.)}$'s are the correlation matrices corresponding to the data and model. The resulting values of the sessions with frozen trials were 0.952 on average (s.e.m. 0.007). The results show that the extracted latent factors captures well the co-variability of the population with only 4 dimensions. To compare with linear models, we performed PCA on the raw spike train with 100 ms bins. S4 Fig shows the cumulative explaining power of the PCs. The $R^2$ of the first 4 PCs are −0.877 on average (s.e.m. 0.303). The negative predictive $R^2$ values show that 4 PCs are not adequate to capture the shared variability of neural activity.

## Stimulus-encoding is concentrated in one shared dimension

In previous work, MT neurons showed strong transient responses on average to motion pulses [27]. We ask if the individual MT responses to visual stimulus are aligned at the population level. To describe the temporal dependence of the latent factors on the motion pulses, we calculated the pulse-triggered average (PTA) for each of the seven pulses [27]. The PTAs are the regression coefficients that predict the change in latent states. Specifically, each PTA corresponding to one of the 7 motion pulses represents the modulation of latent factors by a unit visual motion (a single patch of Gabor drifting in one direction during a pulse), assuming a linear scaling with motion strength (see Materials and methods).

The latent factors are subject to arbitrary rotation [26] which results in models with equivalent explanatory power. Hence, we rotated the latent factors for each session so that the effects of motion pulses are concentrated in decreasing order across dimensions (Fig 4A). For both subjects, the pulses are faithfully represented as transiently modulated latent factors, and most of the motion information is encoded in the mean value of the first factor—we refer to this factor as the *stimulus axis*.

We pooled the stimulus-explaining latent factors alignment across all sessions. The first dimension explains most (> 90%) of the PTA in the latent factors for all but one session (Fig 4B). This concentration of stimulus information in 1-dimension is consistent with the canonical view of MT as primarily a sensory area. Since the sensory stimulus is 1-dimensional (directional motion with different strength), this suggests that the encoding of MT units is temporarily uniform (without multiple time scales of adaptation or lag) and linear (no nonlinear superposition). Note that this is not a trivial result, since the motion information can be encoded in a curved 1-dimensional manifold that spans multiple dimensions in the neural space [35].

## Sensory and choice population codes are misaligned in MT

Next, we investigate how the downstream choice signal is aligned with respect to the stimulus axis. There are several possibilities that the choice-correlations can manifest in the MT population activity (Fig 1). To optimally perform the task, the choice should rely only on the stimulus

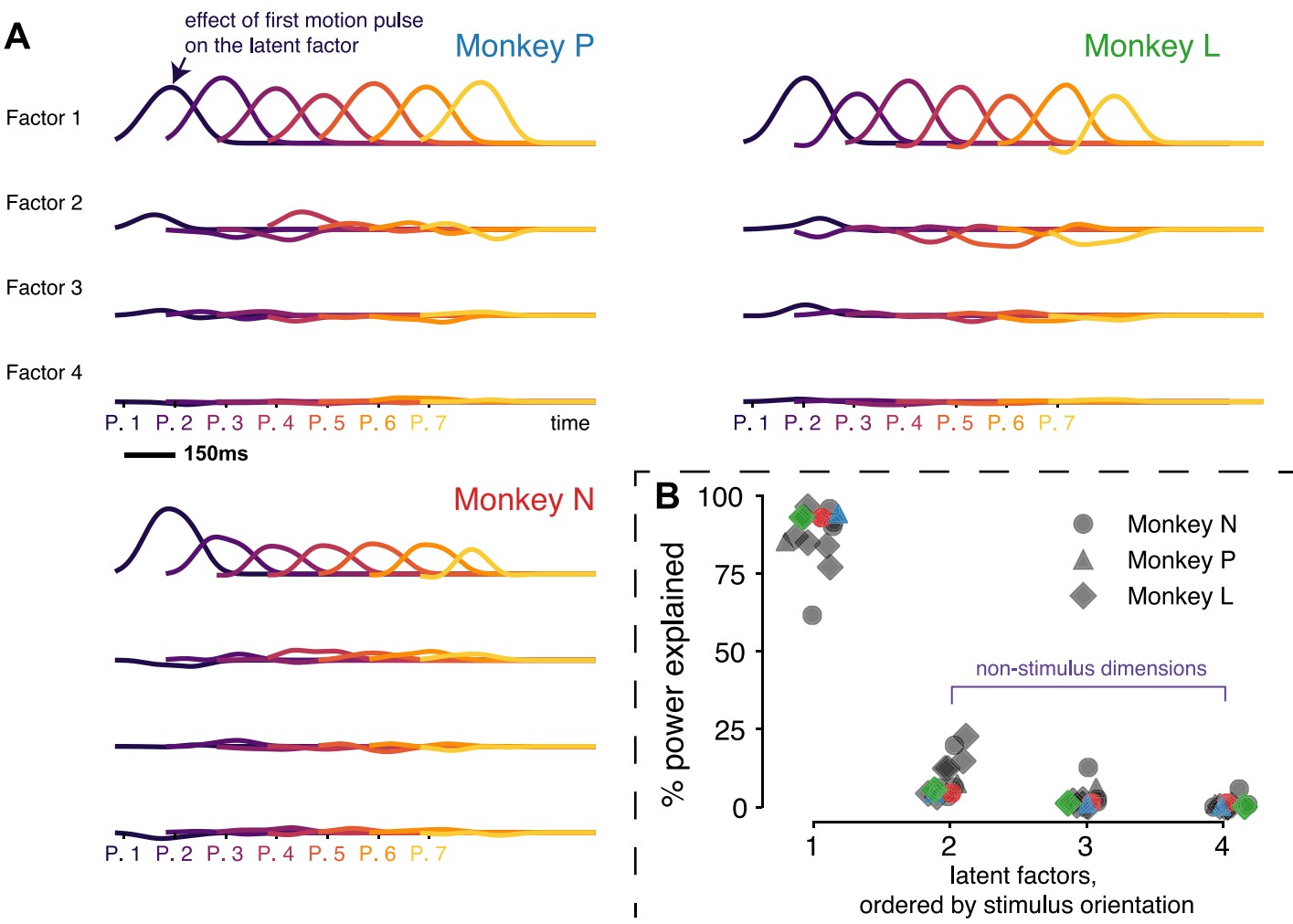

**Fig 4. Visual motion pulse information encoded in one dimension.** (A) Pulse-triggered average of three example sessions, one from each monkey, are shown. The factors are rotated such that most of the stimulus power is in the first factor. They visualize the weights of pulses on the latent factors that were estimated from respective sessions. The color gradient indicates the seven pulses of visual motion stimuli. Each pulse last 150ms. (B) The power of each factor that explains the variation contributed by the stimuli to the factors. Each marker indicates one session, the shape indicates the animal and the color indicate the respective example sessions in (A).

and ignore the off-axis "noise" [17]. Hence, for a purely feed-forward system, only the noise in the stimulus dimension should influence the choice, resulting in choice-correlation reflecting the optimal strategy (Fig 1C). Otherwise sub-optimal "readout" can show choice-correlation through stimulus-irrelevant variability (Fig 1D). On the other hand, feedback paths can mix the downstream choice process signals back into the MT representation: if the feedback is aligned with the stimulus-axis, it will corrupt the encoding of the sensory signal (Fig 1E), while misaligned feedback that stays orthogonal to the continuous stream of stimulus modulated population activity subspace (Fig 1F).

To investigate the effect of different axes on the choice, we calculated the choice probability of the recorded neural population after mapping the multivariate activity to choice through *choice-mapping* (Fig 5; see Materials and methods). The pooled choice probability estimated using the choice-mapped stimulus-axis, non-stimulus-axes (the 3-dimensional subspace orthogonal to the stimulus-axis), and all 4 dimensions of the MT latent factors are 0.546 (s.e. m. across sessions: Monkey N 0.013, Monkey P 0.007 and Monkey L 0.012), 0.591 (s.e.m.

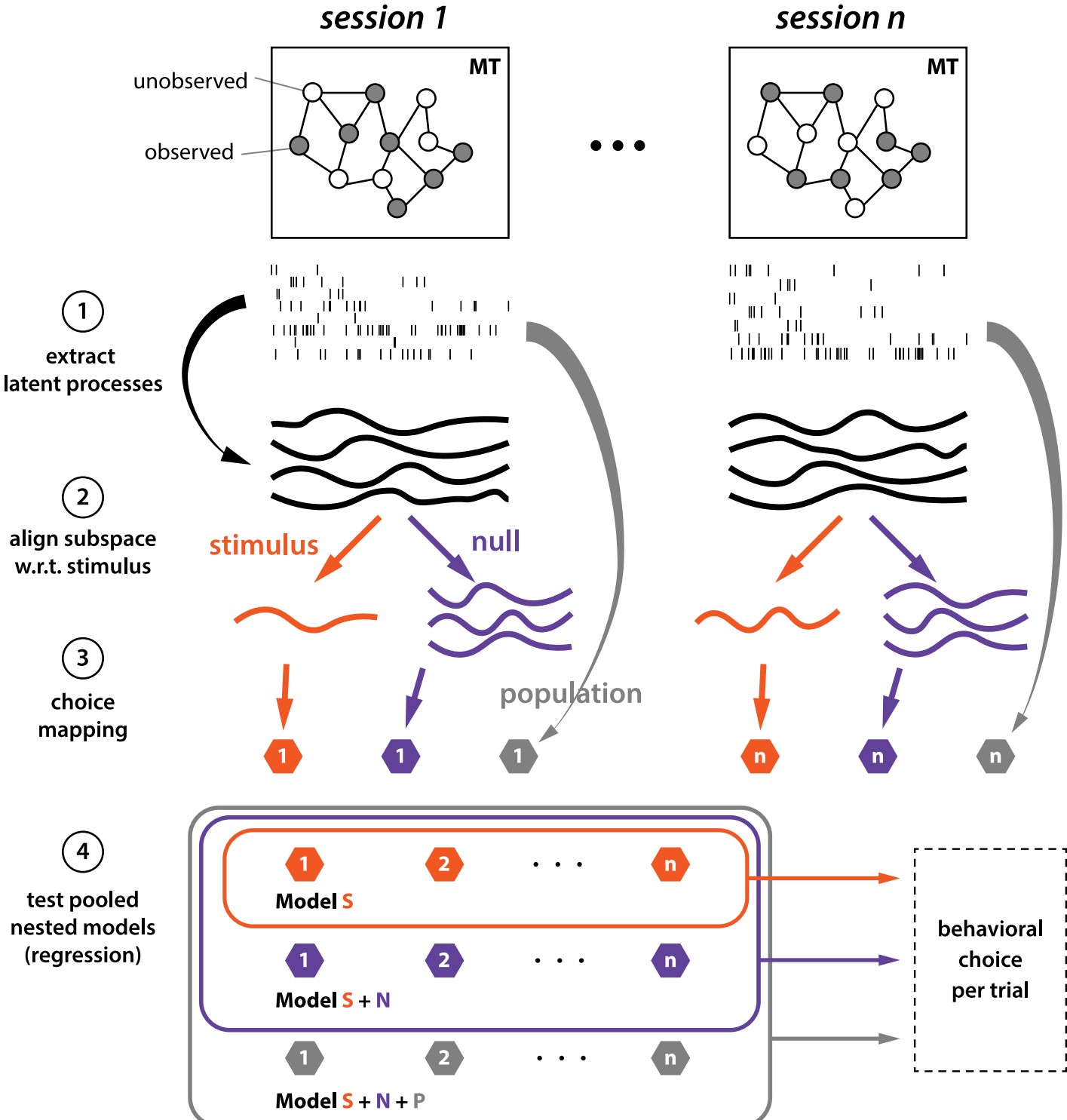

**Fig 5. Data analysis pipeline and nested model comparison.** (1) Extract latent factors. (2) Align latent factors to stimulus & null dimensions. (3) Map dimensions of latent factors into real-valued scalars. (4) Pool the choice-mapping over all sessions and perform nested log-likelihood tests.

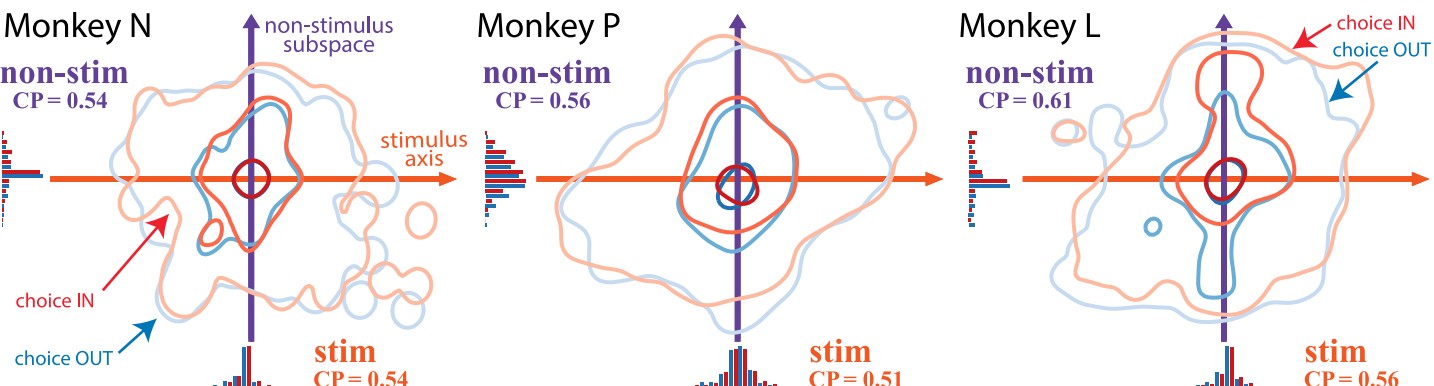

**Fig 6. Choice probabilities of latent factors for each monkey.** Contours corresponds to 50%, 90%, 99% quantities of the choice-mapped stimulus and non-stimulus trial distribution. The IN choice distribution (red-shade contours) is biased upward, indicating existence of the choice information in the non-stimulus axes. The pooled choice probability estimated using the choice-mapped stimulus-axis, non-stimulus-axes (the 3-dimensional subspace orthogonal to the stimulus-axis), and all 4 dimensions of the latent factors are 0.546, 0.591, and 0.621 respectively. The estimated population spike count choice probability is 0.627. For nested statistical tests of the corresponding regression models, see main text and Fig 5.

across sessions: Monkey N 0.009, Monkey P 0.020 and Monkey L 0.027), and 0.621 (s.e.m. across sessions: Monkey N 0.013, Monkey P 0.012, Monkey L 0.011) respectively (Fig 6). The estimated population spike count choice probability is 0.627. To verify that the pooling across sessions (Fig 5, stage 4) does not weaken the choice information, we compare the pooled model with models of individual sessions as a baseline. The likelihood ratios of full models of individual sessions to the full model of pooled sessions is between 0 and 1 because the log-likelihoods are always negative and the pooled model is at most as good as the individual session models. Among the 10 sessions, the likelihood ratio ranges between 0.92 and 0.99, and the average is 0.98. These indicate that the pooling over sessions keeps most of the choice information. We confirmed that the higher dimensionality of the non-stimulus subspace does not result in inflating choice probability (see Material and methods, and S1 Fig). We have further verified that the CP on the stimulus axis defined by the logistic regression weights from the raw spike counts to the stimulus is much smaller (stim.: 0.53, nonstim.: 0.61), consistent with the nonlinear latent factor results.

To determine the major contribution among regressors on choice, we performed nested likelihood ratio tests by adding the choice-mapped value of stimulus-axis, non-stimulus-axes, and the population, one by one (Fig 5). The choice is significantly correlated with the latent non-stimulus subspace ($p < 2.2 \times 10^{-16}$), which indicates that the choice axis is not perfectly aligned with the stimulus axis as the optimal readout or corrupting feedback models suggest. Therefore, our analysis supports representation of choice information in the non-stimulus latent subspace. This misalignment of stimulus axis and choice axis can occur through either non-optimal readout (Fig 1D) or non-corrupting feedback (Fig 1F). The misalignment between choice and stimulus in MT provides evidence for a feedback source of choice information in sensory neurons. The presence of CP orthogonal to the stimulus axis suggests that choice information is not just a result of noise on the sensory response, but rather arises from another process altogether.

### Time course of choice probability indicates feedback of decision-making process to MT

The misalignment between choice and stimulus in MT suggests a feedback source for choice-correlated activity, but could still be explained by suboptimal readout. Debates based on

models and arguments in the literature have yet to resolve issue of feedforward versus feedback choice correlations in area MT [16, 20, 36–38]. To disambiguate the two, we investigate the temporal profile of choice probability. Behavioral analysis showed that the sensory information immediately after its presentation has a strong influence in the choice [27]. In turn, one would expect to see choice information early in the population activity. If the choice information is only present late in the trial, then we can conclude that the feedback from the downstream decision-making process is contributing to the misaligned choice information we observed in the previous section.

To investigate the temporal profile of choice correlation in the non-stimulus axes, we calculated time course of CP. We fit 3 linear choice decoders to the latent non-stimulus axes during the early (200–500 ms), middle (600–900 ms) and late (1000–1300 ms) periods, and then used them to decode the whole period with a 100 ms moving window. Fig 7 shows that the middle and late decoders start climbing late during the visual motion presentation and reach a peak at around the motion stimulus was terminated. This temporal profile is consistent with a choice variable that accumulates sensory evidence [12], and supports the non-corrupting feedback from the decision-making process. On the other hand, the early decoder shows a constant choice probability throughout the motion presentation period (Fig 7) which could represent a per-trial choice bias. These observations suggest that the choice information resides in more than one dimension within the non-stimulus subspace.

## Discussion

To understand how stimulus and perceptual choice information is represented across the population of MT neurons, we take advantage of recent developments in unsupervised statistical approaches to single-trial population analyses (Fig 8). We use a Bayesian inference framework, vLGP [26], to demix the visual stimulus, the reported choice, and the trial-to-trial variability signals presented in the population activity. In contrast to other demixing methods [39, 40], our approach does not require trial-averaging to obtain conditional mean firing rates nor a fixed trial structure necessary for the averaging. Analyzing the trial-averaged responses is convenient as simultaneous recording is not necessary, however, it comes with strong assumptions about the neural code: the noise correlation between neurons are ignored and only the differences in the conditional mean firing rates are assumed to carry useful information. Note that our analysis heavily relies on the marginal correlations (shared variations) present in the simultaneous recording. Moreover, only a single-trial analysis can reveal choice probability.

We found that the low-dimensional latent factors capture the majority of the variability present in the population recordings. By linearly aligning the latent factors to the stimulus and behavioral choice, we were able to investigate how stimulus and choice are shared across neurons. Although we assumed a linear readout [41, 42], note that the optimal nonlinear readout may change this interpretation [43]. Within the space spanned by the latent factors, we found that the sensory task variable was primarily captured by a single latent factor, indicating that high-dimensional visual stimulus was represented in a low-dimensional, task-relevant manner across the MT population and time. This stimulus-dimension was only partially explained by the stimulus, suggesting a presence of information-limiting noise [19, 22, 41, 42] that may strongly influence the animal's choice. However, the choice-correlated variability in the population of was mainly captured in latent subspace orthogonal to the stimulus-encoding dimension. This orthogonality suggests that either the downstream decision circuit used suboptimal readout from MT response (Fig 8, path 3) or the feedback from downstream circuit was non-corrupting (Fig 8, path 4). Further analysis of the time course of choice probability revealed a slow and late rise, supporting the feedback mechanism rather than the readout [20, 37]. The

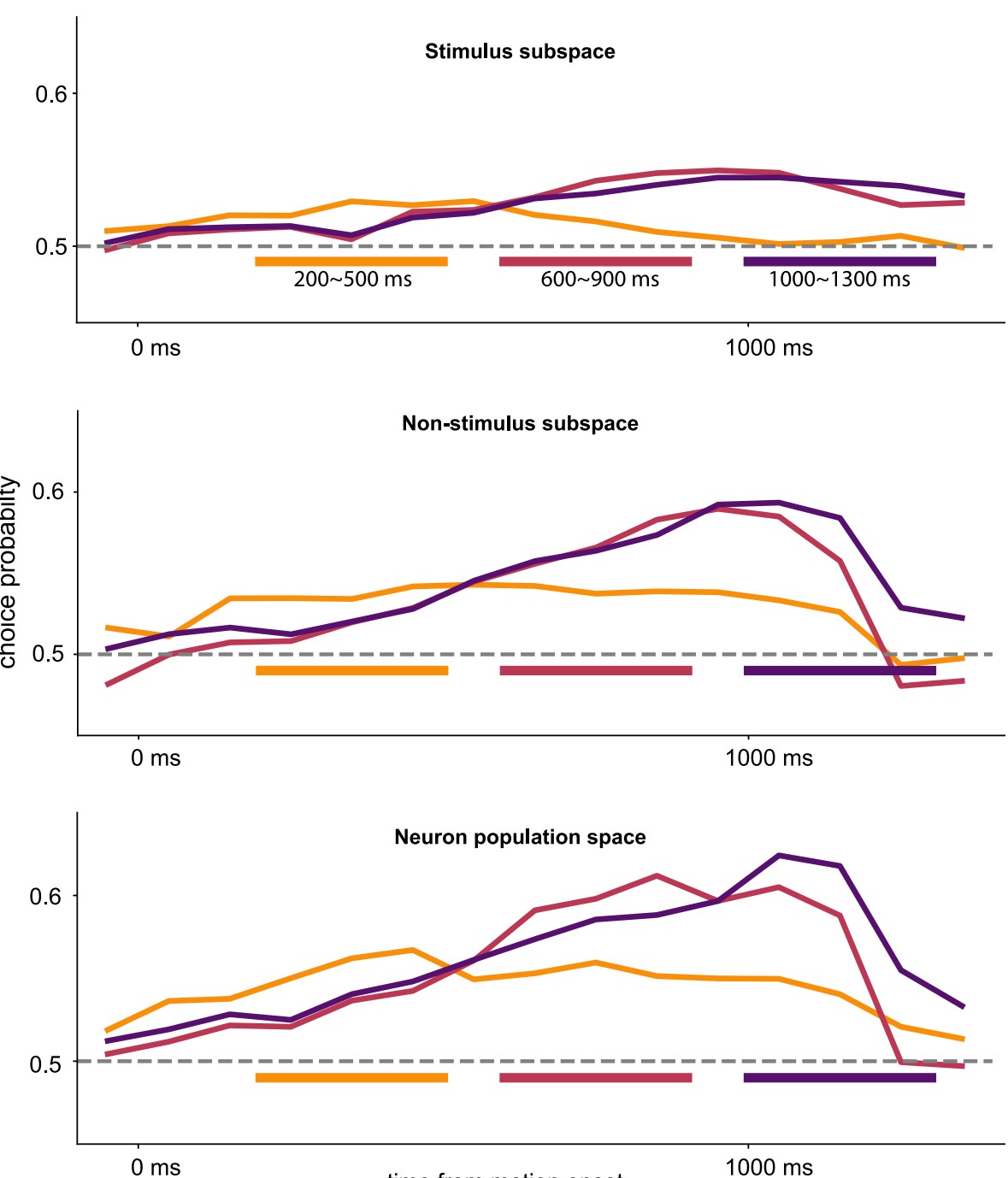

**Fig 7. Time course of choice probability in the latent stimulus subspace, latent non-stimulus subspace and neuron population space suggests feedback from the decision-making process.** Decoders were fit to early (yellow), middle (red), and late (purple) periods (300 ms, marked by the colored bars) of non-stimulus latent factors to predict choice. We used the resulting weights of the decoders to perform choice-mapping on the whole time interval divided into 100 ms non-overlapping moving windows (aligned at the center). The colored curves correspond to the choice probability time course using the respective decoder.

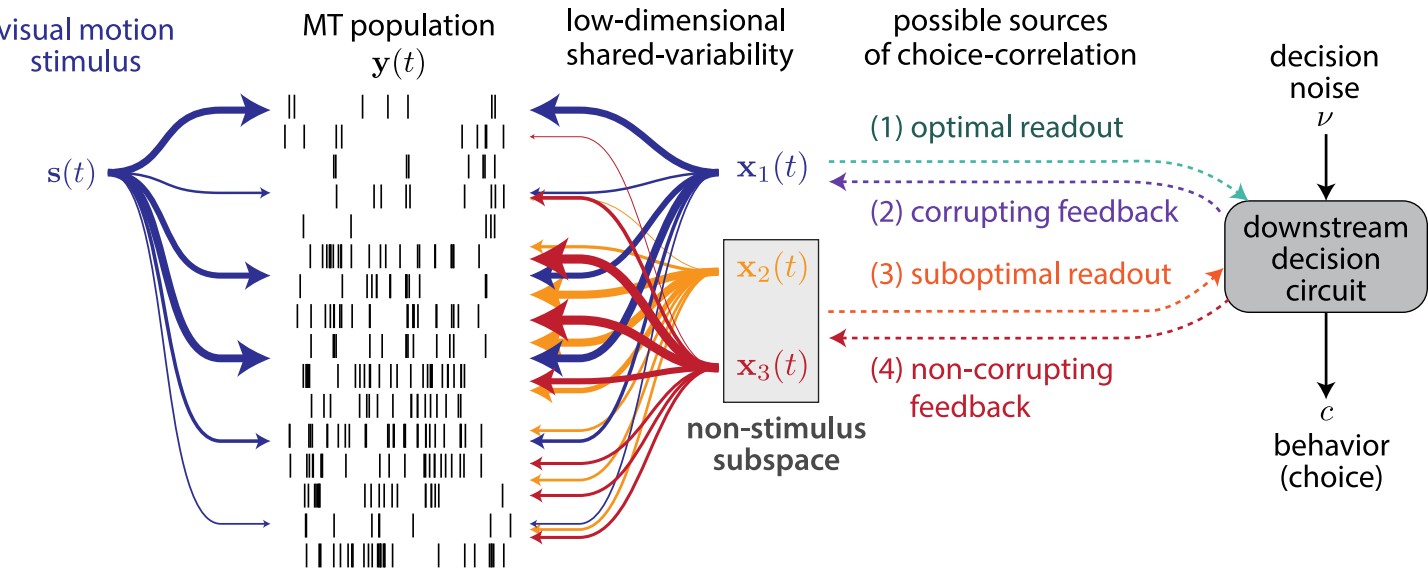

**Fig 8. Four possible sources of neural-choice correlation.** 1-dimensional stimulus drive to MT is picked up as population variability along with other noise correlations denoted $\mathbf{x}_1(t)$, $\mathbf{x}_2(t)$, $\mathbf{x}_3(t)$. To optimally perform the task, the choice should rely on only the stimulus dimension, and hence noise in $\mathbf{x}_1$ shows up as CP in relevant units reflecting their 'readout' strategy (case 1). Non-optimal readout can provide CP through stimulus-irrelevant variability (case 3). Alternatively, feedback from the decision-making process to MT can provide choice-correlation in the stimulus-irrelevant subspace (case 4) without corrupting the optimal representation or the stimulus driven shared dimension (case 2) causing non-optimal behavior.

non-corrupting feedback of choice formation to MT can be useful for tuning of receptive fields and learning of optimal readouts in relation to the task context.

## Supporting information

**S1 Text. Variational latent Gaussian processes.**
(PDF)

**S1 Fig. Choice mapping does not inflate choice probability.**
(PDF)

**S2 Fig. CP (pseudo frozen trial) of latent factors for each monkey.**
(PDF)

**S3 Fig. Time courses of CP (pseudo frozen trials).**
(PDF)

**S4 Fig. Cumulative explaining power of principal components of raw spike trains.**
(PDF)

**S5 Fig. Visual motion pulse information encoded in one dimension of raw spike trains.**
(PDF)

**S6 Fig. Time course of choice probability in the stimulus subspace of raw spike trains.**
(PDF)

## Acknowledgments

We thank the anonymous reviewers for their helpful comments. Memming thanks Hendrikje Nienborg for stimulating discussions.

## Author Contributions

**Conceptualization:** Yuan Zhao, Jacob L. Yates, Alexander C. Huk, Il Memming Park.

**Data curation:** Jacob L. Yates, Aaron J. Levi.

**Funding acquisition:** Alexander C. Huk, Il Memming Park.

**Investigation:** Il Memming Park.

**Methodology:** Yuan Zhao, Alexander C. Huk, Il Memming Park.

**Project administration:** Alexander C. Huk, Il Memming Park.

**Resources:** Alexander C. Huk, Il Memming Park.

**Software:** Yuan Zhao.

**Supervision:** Alexander C. Huk, Il Memming Park.

**Validation:** Alexander C. Huk, Il Memming Park.

**Visualization:** Yuan Zhao, Jacob L. Yates, Il Memming Park.

**Writing – original draft:** Yuan Zhao, Jacob L. Yates, Il Memming Park.

**Writing – review & editing:** Yuan Zhao, Jacob L. Yates, Aaron J. Levi, Alexander C. Huk, Il Memming Park.

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
