## [Decision Letter · Decision Letter 0]

21 Jan 2020

Dear Dr. Park,

Thank you very much for submitting your manuscript "Stimulus-choice (mis)alignment in primate MT cortex" for consideration at PLOS Computational Biology.

As with all papers reviewed by the journal, your manuscript was reviewed by members of the editorial board and by several independent reviewers. In light of the reviews (below this email), we would like to invite the resubmission of a significantly-revised version that takes into account the reviewers' comments.

We cannot make any decision about publication until we have seen the revised manuscript and your response to the reviewers' comments. Your revised manuscript is also likely to be sent to reviewers for further evaluation.

Sincerely,

Daniele Marinazzo

Deputy Editor

PLOS Computational Biology

Reviewer's Responses to Questions

**Comments to the Authors:**

Reviewer #1: The authors applied a nonlinear statistical method to extract latent factors from MT neurons during a decision making task. Then, they analyzed the dimensionality of the stimulus and choice encoding axes. Overal, I found the paper to be interesting and general sound. I have some technical concerns, as well as some confusing aspects of the figures. These concerns should be readily addressable.

1. A major claim of this study relates to feedback signals and choice. However the timeline of the task is not clear to me. Please provide a schematic that shows the timeframe of when the stimulus appears, each of the 7 motion pulses, the subject's response, and when the putative feedback signal is detected. Please also indicate in what time frames the spiking data is analyzed. On pg 11, line 300 you state that you decode using time windows up to 1300ms, but I dont have any reference for what those times mean. Additionally in Figure 3, the horizontal axis is labeled with "time" but no scale is ever given.

2. Fig2. The noise correlation matricies are confusing to me. Is each matrix for a single session? There are 2 sessions for monkey N, and 2 for monkey P, and 7 for monkey L? What is the point of this panel? Is it to show that the model captures the noise correlations? If so, please quantify it instead of visual comparison. If the point is that the noise correlations differ between sessions, again, please quantify it. With respect to concern (1), the timeframe for the latent trajectory goes out to 1500 ms, how does this relate to task structure?

3. (Minor) Fig3. I believe you plotted the factors from three trials, one from each monkey. But the wording of the legend was confusing to me. You might want to change "Three example trials from each monkey are shown" to "Three example trials, one from each monkey, are shown." Again here, what is the scale of time?

4. (Minor) Fig 3. Bottom Right, the label says "stimulus oriented factors" but I think it would be more accurate to say "latent factors, ordered by stimulus orientation" or something like that.

5. (Minor) Fig 3. What does each marker from each monkey mean? Is it the average across all trials from that session? Please clarify.

5. Fig 4. The text states "the IN choice distribtuion ... is biased upwards." Really? For Monkey L that looks true, but not Monkey N. Your main point is that the CPs on the non-stim axis are > 0.5. In your computations of the CPs for each set of subspaces, please report the variability across sessions and across monkeys.

6. Fig 4. How much does your result depend on your nonlinear encoding model? For example, could you arrive at the same conclusion by using a linear method to find the stimulus encoding dimension and computing the choice probability in that dimension vs other dimensions of the data? Please comment on the use of a nonlinear method, or provide a comparison with a linear method.

7. I believe you included "weak stimulus" trials so you could have more data. When you did this you controlled the accuracy rate to be below 65%. I believe this means you did not match the number of hits and misses for these weak stimulus trials. Is it possible that this introduces a correlation between stimulus and choice? Please comment, or ideally, repeat the analysis by randomly removing hit trials until the accuracy rate is 50% (which would match the behavior on zero coherence trials).

8. Please compare your methods and results for this recent paper, which appears to develop a similar method and comes to similar conclusions. https://www.biorxiv.org/content/10.1101/808584v1.full.pdf

Reviewer #2: The authors find a low-dimensional representation of population recordings in MT while animals performed a sensory discrimination task. This low-dimensional representation defines the “stimulus-axis”. They also define different choice axis and find that those orthogonal to the stimulus-axis have a stronger correlation with animals’ choice than the one defined by the stimulus-axis. They claim that this finding provides evidence in favor of sub-optimal read-out with feedback as the origin of choice-related signals in sensory areas. This could be a useful strategy for transmitting feedback signals without harming the neural code.

The manuscript is an interesting piece of work and it represents a relevant contribution to the field. However, the clarity of the introduction, presented results, and methods, is not high and the accessibility could be improved in some sections. I have also additional concerns about specific points in the methodology and results and how they support some of the claimed conclusions, see below.

Major

1. The methods used to address the main scientific question of this study are sound and suited for it. However, a simpler and more straightforward approach to this question should be explored. The authors should compare the direction of two classifiers: one that has been trained on predicting stimulus identity and another that has been trained on predicting animal’s choice. Ideally, the choice classifier should be trained on non-evidence trials (or weak stimulus conditions where stimulus-driven activity has been regressed out). Finding similar results with this simpler analysis, would imply strong evidence in favor of the main finding of this study.

2. When describing Fig. 1 in the Introduction, the authors say (line 42): “In this space, ...”. This statement is slightly misleading because they are neglecting the effect of the covariance matrix on the optimal read-out. If the population’s covariance is not isotropic, the optimal read-out is (Gaussian approximation) proportional to the inverse covariance matrix times the “stimulus axis”. This happens throughout the text, and given that this is central to the motivation of this work, the authors should clarify why they refer to the optimal read-out as parallel to the stimulus-axis and modify the text and figures accordingly.

3. The authors should clarify why the stimulus axis is defined upon the first variable of the 4D latent model. Couldn’t it also be defined directly as the direction joining the mean of the two distributions? Or even better, the direction defined by a classifier trained on decoding stimulus identity?

4. The order the different panels of Fig. 1 appear in the text (Introduction) is a bit chaotic. It is also a bit unclear what are exactly the different hypothesis. Even though in the “Sensory and choice …” section it is mentioned again, the authors should be more explicit about the different hypothesis (i.e. i) …, ii) …, iii) …, etc.) and how they are related to Fig. 1 already in the Introduction. The panels should also ordered in order of appearance in this section.

5. The latent model validation seems to have been performed poorly. In Fig. 2B it is only visually shown how the structure of neuronal correlations is similar to those generated by the latent model. The authors should add a figure or table with the cross-validated explained variance of the model for all monkeys. Fig. 2B should also include a quantification of the similarity between the two matrices.

6. It is unclear why the number of latent variables is set to 4. The authors should specify why their criterion was based on the session with the largest number of neurons. Is it consistent across all monkeys? Is 4 the optimal number for only that session, for few of them, or for most of them?

7. Comparing the influence of the stimulus-axis (1-dimensional) and the non-stimulus space (3-dimensional) is somehow unfair because 3 dimensions will in general be more informative than 1 dimension. It would be strong evidence in favor of the hypothesis of this study to show that the CP for the stimulus axis is smaller than the CP for each non-stimulus dimension evaluated individually. The authors should address this point, and in case of a negative result, explain how this can be consistent with their hypothesis.

8. Fig. 6 is confusing. Why is the stimulus-axis CP (time profile) not shown in this figure? The fact that curves purple and red peak later in the trial is expected because of the way the classifiers were trained. I am a bit skeptical that Fig. 6 shows indeed evidence in favor of the main finding of this study. Using these curves as well as the stimulus-axis CP time profile, the authors should be more clear on why Fig. 6 is provides evidence in this regard.

Minor

- In Fig. 1 it would be helpful to also depict how information is affected for each different scenario (panel). Please add some graphical clarification in this regard.

- Please, add a paragraph where it is briefly explained how the feedback signals can affect CP. Readers might not be fully familiar with that literature and it would make the text easier to follow.

- Even though the behavioral protocol is explained in ref. 24 and 29, it would improve the clarity and readability of the manuscript to describe the experiment in more detail in the methods section. The reader should be able to get a good picture of the dataset without the need to read refs. 24 and 29. For instance, the “Pulse-triggered average” section in the methods would be clearer with more detailed description of the stimulus presented to the monkeys.

- In the Methods section some of the equations are numbered and others are not. It would improve the clarity of the manuscript to have all numbered.

- In the “Single-trial latent dynamics of population” section (Methods), the vLGP model is poorly explained. For readers that are unfamiliar with variational inference it would be useful to have an explicit expression for the loglikelihood of the model as well as the ELBO term to be maximized. It should also be stated what term in the ELBO is going to be approximated by the equation in line 112.

- There is a typo in line 118: “the” shouldn’t be there.

- Labels and figures (everything) in Fig. 2 are too small and therefore very difficult to read. They should be made larger.

- The “Pulse-triggered average” section is unclear. It should be re-written to make it more easier to follow. In line 140, Beta hasn’t been defined (is it a typo for W?).

- The term “frozen trials” is important for this study and is used throughout the manuscript. However, it is never properly defined. Please, add a sentence both in the methods and the results section defining the term.

- The last section of the Methods should be rewritten in a clearer way. Even tough in the results section it is visited again, it is conceptually a fundamental part of the study and it should be more explicitly described already in the Methods. In particular, it is not well stated how are c_stimulus and c_nonstimulus calculated.

- Fig. 4 is difficult to understand. The authors should add a more detailed explanation on both the results section and the Figure caption. Specifically they should make sure they answer: what is the distribution plotted for each monkey? How does this figure relate to Fig. 1? How does each panel relate graphically to its shown CP? Also, text is difficult to read, the size should be increased.

- For point 2 of the major revisions, Averbeck et al. Nat. Rev. Neuro. (2006) and Nogueira et al., J. of Neuro (2019) could be cited given their relevance to this topic.

- In line 15, Kafashan et al. (2020) bioRxiv2020.01.10.902171; and Bartolo et al. J. of Neurosci. (2020) 2072-19 are also very relevant studies in this regard.

**Have all data underlying the figures and results presented in the manuscript been provided?**

Reviewer #1: Yes

Reviewer #2: Yes

PLOS authors have the option to publish the peer review history of their article (what does this mean?). If published, this will include your full peer review and any attached files.

Reviewer #1: No

Reviewer #2: No
---

## [Decision Letter · Decision Letter 1]

5 Apr 2020

Dear Dr. Park,

We are pleased to inform you that your manuscript 'Stimulus-choice (mis)alignment in primate MT cortex' has been provisionally accepted for publication in PLOS Computational Biology.

At the same time, please make sure to address the few style recommendations suggested by the reviewers, which you can find below.

Best regards,

Daniele Marinazzo

Deputy Editor

PLOS Computational Biology

Daniele Marinazzo

Deputy Editor

PLOS Computational Biology

Reviewer's Responses to Questions

**Comments to the Authors:**

Reviewer #1: I thank the authors for their response and updates to the manuscript. The methods and conclusion of the paper are more clear, and are sound. I support publication of this manuscript.

One minor change is that they use the abbreviation for area MT without defining it first in the abstract, although they do say "middle temporal area", they should change it to "middle temporal (MT) area".

Reviewer #2: The authors have thoroughly revised the manuscript and have increased significantly the clarity of their main message. It is easier to follow for a broader audience and it fits well within the existing literature. I have been answered most of my concerns with additional analysis and figures that make this study stronger and more robust.

Minor:

- Sentence 413 – 417 is difficult to follow. I would rewrite it or split it in two.

- Fig. 6 is mentioned in the text before Fig. 5 (line 328).

- y axis in Fig. S1 seems too short on the upper part.

- Text in Fig. S2 difficult to read due to small font.

- y label is missing in Fig. S3 and S6.

**Have all data underlying the figures and results presented in the manuscript been provided?**

Reviewer #1: Yes

Reviewer #2: Yes

PLOS authors have the option to publish the peer review history of their article (what does this mean?). If published, this will include your full peer review and any attached files.

Reviewer #1: No

Reviewer #2: No

---

## [Editor Report · Acceptance letter]

11 May 2020

PCOMPBIOL-D-19-02195R1 

Stimulus-choice (mis)alignment in primate area MT

Dear Dr Park,

I am pleased to inform you that your manuscript has been formally accepted for publication in PLOS Computational Biology. Your manuscript is now with our production department and you will be notified of the publication date in due course.

With kind regards,

Laura Mallard
